# Multivesicular Liposomes for the Sustained Release of Angiotensin I-Converting Enzyme (ACE) Inhibitory Peptides from Peanuts: Design, Characterization, and In Vitro Evaluation

**DOI:** 10.3390/molecules24091746

**Published:** 2019-05-05

**Authors:** Ning Li, Aimin Shi, Qiang Wang, Guoquan Zhang

**Affiliations:** 1College of Food Science and Engineering, Northwest A & F University, Yangling 712100, China; sy76534371@126.com; 2Institute of Food Science and Technology, Chinese Academy of Agricultural Sciences/Key Laboratory of Agro-Products Processing, Ministry of Agriculture, P.O. Box 5109, Beijing 100193, China; sam_0912@163.com

**Keywords:** peanut peptide, multivesicular liposomes, ACE, controlled-release, digestive stability

## Abstract

The multivesicular liposome (MVL) provides a potential delivery approach to avoid the destruction of the structure of drugs by digestive enzymes of the oral cavity and gastrointestinal system. It also serves as a sustained-release drug delivery system. In this study, we aimed to incorporate a water-soluble substance into MVLs to enhance sustained release, prevent the destruction of drugs, and to expound the function of different components and their mechanism. MVLs were prepared using the spherical packing model. The morphology, structure, size distribution, and zeta potential of MVLs were examined using an optical microscope (OM), confocal microscopy (CLSM), transmission electron cryomicroscope (cryo-EM) micrograph, a Master Sizer 2000, and a zeta sizer, respectively. The digestion experiment was conducted using a bionic mouse digestive system model in vitro. An in vitro release and releasing mechanism were investigated using a dialysis method. The average particle size, polydispersity index, zeta potential, and encapsulation efficiency are 47.6 nm, 1.880, −70.5 ± 2.88 mV, and 82.00 ± 0.25%, respectively. The studies on the controlled release in vitro shows that MVLs have excellent controlled release and outstanding thermal stability. The angiotensin I-converting enzyme (ACE) inhibitory activity of ACE-inhibitory peptide (AP)-MVLs decreased only 2.84% after oral administration, and ACE inhibitory activity decreased by 5.03% after passing through the stomach. Therefore, it could serve as a promising sustained-release drug delivery system.

## 1. Introduction

High blood pressure is one of the leading risk factors for global mortality and is estimated to have caused 9.4 million deaths and 7% of disease burden, as measured in disability-adjusted life years (DALYs) [1]. The global prevalence of high blood pressure (defined as a systolic and/or diastolic blood pressure ≥140 and 90 mmHg, respectively) in adults aged 18 years and over was around 22% in 2014 [2]. The proportion of the world’s population with high blood pressure or uncontrolled hypertension fell modestly between 1980 and 2010. However, because of population growth and ageing, the number of people with uncontrolled hypertension has risen over the years. High blood pressure is a major cardiovascular risk factor [3]. If left uncontrolled, hypertension causes stroke, myocardial infarction, cardiac failure, dementia, renal failure, and blindness. These disease conditions cause human suffering and impose severe financial and service burdens on health systems [4].

Bioactive peptides are protein fragments, which are inactive in a precursor sequence. After their release, they interact with receptors in the body and regulate the function of particular systems, such as nutrient uptake and immune defense. They can also act as carriers of metal ions, opioids, antioxidants, antimicrobials, anti-adhesives, anticancer, and angiotensin I converting enzyme (ACE)-inhibitory drugs [5,6].

Bioactive peptides, generated from peanut meal by enzymatic hydrolysis, have been shown to enhance functional properties [7] and improve nutritional quality, as well as raise added value [8]. In particular, peanut protein hydrolysates are known to be a good source of ACE-inhibitory peptides [8,9,10], which indicates they may be beneficial for blood pressure regulation. However, there is limited information on suitable vehicles intended to protect these peptides from hydrolysis by gastrointestinal digestive enzymes, as well as on the sustained delivery of ACE peptides from peanuts.

Some dosage forms have been introduced, such as liposomes [11], nanoliposomes [12], nanoparticle suspension [13], oral microemulsion [14,15], and solid dispersions. However, these showed some shortcomings, such as poor long-term stability [16], small drug loading [17], and unobvious release effect. Multivesicular liposomes (MVLs) have received great attention due to their excellent stability and longer duration of release [16,18].

The objectives of this investigation were to assess (1) the ACE-inhibitory activity of enzymatic hydrolysates obtained from the by-product of peanut meal, (2) the multivesicular liposome formulations for the sustained delivery of peanut peptides: preparation, and (3) their characterization (Figure 1).

## 2. Results

### 2.1. Characterization of Angiotensin I-Converting Enzyme (ACE)-Inhibitory Peptide (AP) and ACE Inhibitory Activity

The enzymatic hydrolysis of protein by-products leads to value-added hydrolysates with improved biological activity, which can be used as a functional ingredient in food, nutraceuticals, and pharmacological formulations. The effect of proteolytic enzyme modification on the ACE-inhibitory activity of peanut peptides was evaluated.

The progress of hydrolysis was monitored by determining the degree of hydrolysis (DH) (%) and TCA-soluble nitrogen index TCA-NSI (%). DH and TCA-NSI are important factors controlling the composition and biological properties of the obtained hydrolysates. In our previous research, the enzymes and the main factors were optimized by single factor experiment. Neutrase and Protamex were the best enzymes of those tested. The optimal conditions were obtained as follows: a ratio of solid to liquid of 1:8; an enzyme quantity of neutrase and protamex of 5200 and 374.4 U/g, respectively; an enzymatic temperature is 40 °C; and an enzyme digestion time of 2.0 h for hydrolyzed neutrase and 2.0 h for hydroluzed protamex sequentially. Under these conditions, a DH of 41.40 ± 0.53% and a TCA-NSI of 85.25 ± 0.32% was achieved.

Beyond the fine structure of bioactive peptides, molecular size has also been identified as an important determinant of their ACE-inhibitory activity, solubility, and some other functional characteristics. Our research shows that liquid hydrolysate primarily contained peptides <1000 Da in size. To compare our hydrolysates with others reported in literature, the molecular weight of <1000 Da accounted for 95% of the hydrolysate, which was higher than the proportion reported previously.

ACE catalyzes two main reactions responsible for the constriction of blood vessels that leads to blood pressure elevation. Its IC_50_ values ranged from 0.126–246.7 mg/mL, as reported previously, and a value of 0.77 mg/mL was obtained in this study [19].

### 2.2. Effect of Technology Process Parameters on the Particle Encapsulation Efficiency of Angiotensin I-Converting Enzyme (ACE)-Inhibitory Peptide Multivesicular Liposomes (AP-MVLs)

In the study of the preparations of angiotensin I-converting enzyme (ACE)-inhibitory peptide multivesicular liposomes (AP-MVLs), nine factors (namely first-vortex, first-vortex time, second-vortex time, lecithin concentration, cholesterol concentration, triolein concentration, l-lysine concentration, the ratio of Oil/Water(O/W)(mol/mol), and volume ratio of primary emulsion/second aqueous solution) were investigated. Five levels were designed for each factor in the single-factor experiment, as shown in Table 1 and Table 2.

Figure 1 and Figure 2 exhibit the results of the single-factor experiment. The data were pooled from three independent trials.

As shown in Figure 1, when the rotational speed was increased from 12,000 to 25,000 rpm, the encapsulation efficiency of AP-MVLs increased from 71 ± 1.25 to 82.05 ± 2.38%. Within the range of 12,000–18,000 rpm, the encapsulation efficiency of the MVLs was between 71.03 and 74.8%, and the speed had little effect on the encapsulation efficiency. The initial emulsion size of a liposome is mainly influenced by emulsifying speed. When the emulsification condition was more severe, the small droplet of the drug formed was increased, and the liposomes of the polycystic structure can be formed more easily. It is easier to form a polycystic structure of liposomes and increase encapsulation efficiency. In the end, 20,000 was converted to the optimal initial emulsification speed, i.e., when the encapsulation efficiency was 82.00 + 0.25%.

The emulsifying time was within the range of 5–25 min. The encapsulation efficiency was between 72.3 ± 1.25 and 78.91 ± 5.26%. It is helpful to improve the encapsulation efficiency of the loaded substance, but the effect of the emulsifying time on the encapsulation efficiency was limited when the initial emulsification time increased. Using 10 min as the optimal initial emulsification time, at this point, the encapsulation efficiency was 78.91 ± 5.26%.

When the range of the second emulsification time was 5–60 s, the encapsulation efficiency increased from 64.19 ± 3.25 to 73.22 ± 2.58%, and the encapsulation efficiency of liposome changed little after increasing the emulsifying time. Therefore, the optimal emulsification time was 10 s, and the encapsulation efficiency was 73.22 ± 2.58%.

As shown in Figure 2, when the concentration of lecithin changed within the range of 10–20 mg/mL, liposome had little effect on the encapsulation efficiency of the drug. Lecithin was the main membrane in the preparation of liposomes and was a good emulsifier. The addition of adequate amounts of phospholipids helped to improve the encapsulation efficiency of drugs. However, when the concentration of phospholipids was too high, it could form many lipid fragments.

When the concentration of triglycerides increased from 6 to 12 mg/mL, encapsulation efficiency increased slightly. There was little effect on the encapsulation efficiency of liposomes when the concentration of triglycerides changed within the range of 6–16 mg/mL.

When the cholesterol concentration increased from 6 to 12 mg/mL, the encapsulation efficiency increased from 70% to 76%, and the encapsulation efficiency of liposome decreased slightly when the cholesterol concentration increased to 16 mg/mL. The reason for this may be that adding cholesterol, as a “membrane fluidity buffer”, can significantly increase the stability of the thylakoid membrane, reduce drug leakage during the preparation process, and improve drug encapsulation efficiency.

When the O/W volume is 1:2, a very small number of liposomes can be obtained, resulting in many small droplets which cannot be deposited at the bottom of the centrifuge tube. When the volume changes between 1:1 and 3:1, the encapsulation efficiency of the mass is between 75% and 69%, and the decreasing trend of the O/W volume ratio increases.

With the increase of the second aqueous solution, the encapsulation efficiency of liposome decreased, and the primary emulsion/second aqueous solution ratio increased from 1:1 to 1:4. The encapsulation efficiency of the mass was reduced from 74% to 67%. The volume of the second aqueous solution increased, making the drug is more likely to leak. This results in a reduction in the encapsulation efficiency.

When the l-lysine concentration increased from 20 to 40 mmol/L, the encapsulation efficiency increased from 71% to 76%. The encapsulation efficiency was not significantly increased after the amount of l-lysine was increased. l-lysine is a kind of emulsifying agent which has a weak emulsification ability. However, it can enhance the emulsifying ability of phospholipids and can stabilize the recovery of milk by increasing the electrostatic action between the drops.

The preparation process and prescription of liposome were optimized using a single factor experiment, and the preparation conditions were as follows: a soybean lecithin concentration of 16 mg/mL, a cholesterol concentration of 12 mg/mL, a triglyceride concentration of 12 mg/mL, an L-lys concentration of 40 mmol/L, an O/W ratio of 1:1, a primary emulsion/second aqueous solution ratio of 2:3, a colostrum speed of 20,000 rpm, a colostrum time of 10 min, a recovery time of 10 s, an internal water sucrose concentration of 5% (*w*/*v*), and an external water glucose concentration of 7.5% (*w*/*v*). At these conditions, the encapsulation efficiency was 82.00 ± 0.25%, zeta potential was −67.29 ± 3.55, and the sizes were 45–55 μm.

### 2.3. Optical Microscopy (OM), Confocal Microscopy (CLSM), and Transmission Electron Cryomicroscopy (Cryo-EM) in Multivesicular Liposome (MVL) Characterization

Figure 3 shows the morphology of AP-MVLs with an optical microscope, from which we can see that the MVLs were spherical with multiple non-concentric lipid vesicles inside. The opaque parts were the lipid membrane, while the transparent parts were the aqueous rooms.

To study the structure of liposomes by confocal microscopy (CLSM), we loaded the liposomes with a fluorochrome marker which localizes in the lipidic bilayer. CLSM allows us to easily appreciate and evaluate the internal structure of the lipidic systems, which cannot be investigated directly with the other techniques previously described (Figure 3) [20].

In the prescription of multivesicular liposomes, the presence of neutral lipids such as triglycerides is the key component for the formation of stable multivesicular liposomes.

It can be seen in Figure 3 that triglycerides cannot be arranged together with a phospholipid composition polycystic skeleton when organic solvents were removed (over 97%). However, they must stay in the chamber junction and in the water phase so that cavitary fusion in the greater curvature will not occur and, instead, the structure of multivesicular liposomes would be stable [11]. Therefore, lipid content needs to be represented accurately when designing prescription because too little leads to the stability of the structure of multivesicular liposomes and too much leads to the destruction of the structure of the phospholipid bilayer and the confluence of the cavity.

### 2.4. Zeta Potential, Droplet Size, and Distribution

The profile showed a single narrow peak. Grain size tended to be uniform. The size distribution profile of the AP-MVLs showed a single narrow peak with a mean diameter of 47.6 mm, and the particle size of the span is 1.880.

In this study, the zeta potentials of the prepared AP-MVLs were negative, indicating that the AP-MVLs were negatively charged. The zeta potential values were −67.29 ± 3.55 mV. Zeta potentials greater than 30 mV or less than −30 mV can stabilize these double emulsions [21].

### 2.5. In Vitro Controlled Release

#### 2.5.1. Leakage Rate of AP-MVLs under Different Temperatures

MVLs have good encapsulation efficiency for water-soluble peptides like liraglutide because their water space is more than 95% [18]. MVL particles release peptides in a more uniform manner, unlike a polymer-based microsphere system that exhibits an initial burst release of protein from the matrix.

At room temperature and 4 °C, the leakage rate for 144 h was 21.3 and 59.4%, respectively, and the liposomal drug leakage happened under the two conditions above. As shown in Figure 4, when the liposomes were stored at 4 °C, the leakage rates of liposomes were relatively low, and its tendency to increase with time is slow.

#### 2.5.2. Cumulative Release of AP-MVLs with Different Usages of Cholesterol and Trioleate

As shown in Figure 5, the anti-hypertensive peptides from peanuts can be slowly released from MVLs. The dosages of cholesterol were 4, 12, and 20 mg/mL respectively. The drug release of MVLs in the first 0.5 h were 23.58, 11.24, and 10.38%, respectively. These were lower than 40%, and there was no obvious sudden release. High (20 mg/mL) or low levels of cholesterol (4 mg/mL) all lead to the release of the drug from the liposomes. When the amount of cholesterol is 4 mg/mL, more than 90% of the antihypertensive peptides are released at about 60 h. At 120 h, the drug release in the 12 and 20 mg/mL groups was 84.24 and 85.91%, respectively, in line with the provisions of the Pharmacopoeia. As mentioned earlier, cholesterol has the effect of regulating the fluidity of the membrane. Adding a proper amount of cholesterol can increase the stability of the lipid membrane and slow the release rate of the drug, while excessive cholesterol may increase the rigidity of the membrane and cause the lipid membrane to be more easily broken.

As shown in Figure 5, liposomes with different amounts of triolein can also achieve the slow release of drugs. At 0.5 h, the release rates of liposomes in the 4, 12, and 20 mg/mL groups were 13.81, 11.24, and 12.85%, respectively, without any sudden release. At 120 h, the cumulative doses of drugs in the three groups were 70.85, 80.35, and 88.58%, respectively. The release rate increased with the increase of triolein. As mentioned earlier, triolein mainly plays the role of bridging, connecting, and stabilizing the internal vesicles of multivesicular liposomes. There are some reports that the length of the carbon chain of triglycerides has a great effect on the release rate of liposomes [17,22]. In the experiment, triolein was the alkyl chain length, and the long chain triglyceride of 18 C had a good release effect. However, the excess of triolein can be found in the water phase in the form of small droplets, which may change the acidity of the internal water phase and destroy the osmotic pressure in the water phase of the lipid in the body. This results in the accelerated release of the drug.

#### 2.5.3. Function Characteristics of AP-MVLs

##### Thermal Stability

Peptides have the characteristics of degradation at high temperature, which seriously restrict the application of polypeptide drugs and health products, and bring difficulties for the production, transportation and storage of peptides [23]. The effects of temperature conditions at 4, 10, 20, 30, and 60 °C on the content and inhibitory activity of polypeptides were investigated in this paper, which provided a theoretical basis for the industrial application of peptides.

As shown in Figure 6, under the temperature conditions of 4, 10, 20, 30 and 60 °C, the content of peanut peptide in the solution decreased with the increase of temperature and the prolongation of heat preservation time. At 4 °C, the degradation was 1.00% and the inhibitory activity of ACE was reduced by 9.35%. Under the condition of heat preservation at 60 °C, the degradation was 45.00%, and the inhibition activity of ACE was reduced by 49.63%.

As shown in Figure 6, the absorbance of multivesicular liposomes at 4 and 37 °C was basically stable at 72 h. This shows that multivesicular liposomes have good stability, and it is feasible to investigate the encapsulation and release of drugs in this experimental environment [24].

##### Digestive Stability

In vivo antihypertensive activity is the final evaluation index of ACE inhibitory polypeptide. When the ACE inhibitory polypeptide enters the human body by oral administration and resists the hydrolysis of digestive enzymes in the oral and gastrointestinal tract, it can enter the blood circulation in a complete peptide chain and eventually reach the target cell. In order to study the anti-digestive performance of polypeptides and provide basic data for the bioavailability of peptides, artificial saliva and artificial gastric juice can be used to simulate the human body’s digestion and absorption system [15].

As shown in Figure 7, the ACE inhibitory activity of peanut peptides decreased by 59.55% after oral administration. After gastric administration, the ACE inhibitory activity of peanut peptides decreased by 67.32%. This shows that peanut peptides have poor stability in simulating the human digestive system.

Simultaneously, the ACE inhibitory activity of peanut peptide multivesicular liposomes decreased only 2.84% after oral administration, and the ACE inhibitory activity decreased by 5.03% after passing through the stomach. It is evident that the multivesicular liposomes still have a high ACE inhibitory activity after passing through the mimic digestive system, which fully shows that the multivesicular liposomes have fully protected the structural integrity of the peanut peptide.

In the simulated digestion test, saliva amylase, pepsin, and trypsin played a key role. Several enzymes catalyzed the breaking of aromatic amino acids, glutamic acid, leucine, and other peptide bonds. Trypsin tends to hydrolyze the peptide bonds formed hydroxyl by lysine and arginine.

In the experiment, the activity of ACE decreased significantly after oral and gastric preparations, indicating that the sequence contained a large number of the above amino acids. Meanwhile, we speculate that the peanut peptide obtained in this experiment is a substrate ACE inhibitor.

## 3. Discussion

We have successfully prepared a novel controlled release drug delivery system made of AP-MVLs. AP-MVLs with high encapsulation were prepared using the double-emulsion method according to the formulation optimized by the central composite design. They showed a remarkable sustained release effect and prolonged circulation time. Furthermore, by studying the physicochemical properties and structural characteristics of the AP-MVLs, the stability of peanut peptides and the AP-MVLs were determined. Then, the thermal stability and digestive stability of the peanut peptide and the AP-MVLs were investigated. The multivesicular liposomes could not avoid the structure of the peanut peptide at high temperatures. The destruction of peanut peptides can destroy the digestive enzymes of the oral cavity and gastrointestinal tract. Our study demonstrated that MVLs could encapsulate water-soluble substances. Therefore, they can provide a new strategy for poorly soluble drugs, and AP-MVLs may serve as a promising sustained-release drug delivery system.

## 4. Materials and Methods

### 4.1. Substrate

Peanut meal (PM) was purchased from the Lan Shan Group, Shandong Province, China. Neutrase from *Bacillus subtilis* (EC No. 3.4.24.28; 60,000 U/g) and Protamex from *B. subtilis* (EC No.3.4.21.62; 120,000 U/g) was purchased from Solarbio Science and Technology Co., Ltd. (Beijing, China). Cholesterol, l-Lysine and phospholipid were purchased from Solarbio Science and Technology Co., Ltd. (Beijing, China). Acetonitrile was chromatographically pure and purchased from Merck & Co., Inc. (Kenilworth, NJ, USA). Hippuryl-histidyl-leucine (HHL) was purchased from Sigma Chemical Co. (St. Louis, MO, USA). All chemicals used in this investigation were analytical grade and purchased from Beijing Chemicals Co. (Beijing, China).

### 4.2. Preparation of ACE-Inhibitory Peptides (AP)

All processing was performed in the Institute of Food Science and Technology, Chinese Academy of Agricultural Sciences or the Key Laboratory of Agro-Products Processing, Ministry of Agriculture. A conical flask was used to prepare all PM dispersions. The PM dispersions were prepared in deionized water (1:4, *w*/*v*). Mixing and dispersions were provided using a magnetic stirrer (IKA Works, Inc. Wilmington, USA)to ensure thorough mixing. Enzymatic hydrolysis was carried out in a stable temperature horizontal shaking bath. The peanut meal(PM) dispersions were equilibrated for 20 min at 45 °C, the optimal temperature for protamex and neutrase hydrolysis. After the 20 min incubation period, the first enzyme was added to the dispersion immediately, and this was hydrolyzed for 120 min at 45 °C. After the 120 min hydrolysis period, the second enzyme were added to the dispersion and hydrolyzed for 120 min at 45 °C. After being hydrolyzed, the hydrolysate was treated in a 90 °C water bath for 10 min to inactivate the enzyme. The decanter(IKA Works, Inc. Wilmington, DE, USA) was set to 4500 r/min for 20 min. The liquid stream from a decanter (liquid hydrolysates) was collected and finally dried into powder using a Büchi mini spray dryer B-290(BUCHI Corporation, New Castle, PA, USA). The co-current drying air flow was set at the maximum rate of approximately 35 m^3^ h^−1^. The inlet temperature (T_in_) was set to 170 °C, and the outlet temperature (T_out_) was maintained at 70 °C by adjusting the feedstock flow rate. Then, the product was collected [25].

### 4.3. Molecular Weight Distribution

Relative molecular weight distributions of hydrolysates were determined by size-exclusion chromatography using a Waters Breeze System (Waters Co., Ltd. Milford, MA, USA). Hydrolysates were separated on a TSK gel 2000 SWXL (300 × 7.8 mm, Tosho Co., Ltd. Numama, Zushishi, Kanagawaken, Japan), pre-equilibrated with aqueous acetonitrile (55:45, *v*/*v*) containing trifluoroacetic acid (0.1 mL/100 mL), and eluted with the same solvent at a flow rate of 0.5 mL/min. Ultraviolet (UV) light absorbance was monitored at 220 nm. A calibration curve was prepared using the following external standards: Cytochrome C (12,500 Da), aprotinin (6511 Da), bacitracin (1422 Da), and glutathione (189 Da), all of which were obtained from Sigma Aldrich (St. Louis, MO, USA).

### 4.4. ACE-Inhibitory Activity

In vitro ACE inhibitory activity was assayed according to the method with slight modifications [26].

The reaction mixture contained 50 μL of peptide and 50 μL of ACE (25 mU/mL) and was pre-incubated for 5 min at 37 °C. Then, 150 μL of substrate (8.3 mM HHL in 0.1 M Tris-HCl buffer containing 500 mM NaCl at pH 8.3) was added, and the sample was further incubated at 37 °C for 60 min. The reaction was stopped by adding 250 μL of 1 M HCl. The released HA was extracted using 1.5 mL of ethyl acetate. The upper layer of the ethyl acetate phase (1 mL) was dried in an 80 °C sand bath. The dried sample was dissolved in 1 mL of DI water and filtered through a 0.45-μm membrane filter.

High-performance liquid chromatography (HPLC) using a SunfireTM-C18 (Waters Co., Ltd. Milford, MA, USA). (150 × 4.6 mm) was used for determining the ACE-inhibitory activity of hydrolysates. The chromatographic separations were pre-equilibrated with aqueous acetonitrile (50:50, *v*/*v*) containing trifluoroacetic acid (0.05 mL/100 mL) and eluted with the same solvent at a flow rate of 0.4 mL/min. UV absorbance was monitored at 228 nm. ACE inhibitory activity was calculated as:ACE inhibitory (%) = [ (a − b)/b] × 100(1)
where a and b are the ACE activity in the presence and absence of the hydrolysate samples, respectively. The IC_50_ value, defined as the concentration of the peptide that inhibits 50% of the ACE activity, was determined from the ACE-inhibitory activity and peptide contents of each sample after regression analysis.

### 4.5. Preparation of ACE-Inhibitory Peptide Multivesicular Liposomes (AP-MVLs)

The first step was to prepare the water-in-oil (W/O) solution by emulsifying the sucrose solution with a mixture of soybean lecithin, cholesterol, triolein, and AP dissolved in chloroform, for which the shearing was at 20,000 rpm/min for 10 min. Next, the preformed W/O solution was further emulsified with an aqueous solution containing glucose and l-lysine to get a water-in-oil-in-water (W/O/W) double emulsion. Finally, the W/O/W solution was subjected to a rotary evaporation to remove chloroform, which resulted in the formation of AP-MVLs. The final products were stored at 4 °C for further use.

### 4.6. Morphology and Particle Size of APMVLs

#### 4.6.1. Zeta Potential

The zeta potentials of the AP-MVLs were determined using a zeta potential analyzer (Zeta sizer Nano, Malvern, UK). The samples were diluted with Milli-Q water prior to analysis. The ratio of sample to water was 0.0309:1. In this process, the samples were added slowly to avoid air bubbles. Each sample was analyzed in triplicate at least.

#### 4.6.2. Droplet Size and Distribution

The droplet sizes and distributions of the AP-MVLs were determined using dynamic light scattering using a Master sizer 2000 (Malvern Instruments, Worcestershire, UK) with a measuring range of 20 nm to 2000 mm. Some optical parameters were adjusted as follows: the refractive index of the dispersed phase was 1.466 and the refractive index of the continuous phase was 1.333. Absorption was set at 0.01. Average sizes are reported as D_3,2_.

(2)
$$\mathrm{Span}\;=\;\frac{\left(D(v,0.9)\;-\;D(v,0.1)\right)}{D}(v,\;0.5)$$


The distribution was expressed in terms of span [4,25], defined where *D*(*v*, 0.1), *D*(*v*, 0.5), and *D*(*v*, 0.9) are standard percentile readings from the analysis. *D*(*v*, 0.1) and *D*(*v*, 0.9) are the sizes of the droplets lying below 10% and 90%, respectively, of the sample. *D*(*v*, 0.5) is the median droplet size, which is stated above as the diameter where half of the size lies below this value.

### 4.7. Determination of Encapsulation Efficiency

Exactly 0.1 mL of the mixture containing 1 mL of an AP-MVL suspension and 1 mL of normal saline was added to 2mL of isopropanol and diluted with 10 mL of de-ionized (DI) water. The total content of drug (D_tot_) in the preparations was determined using HPLC. The free drug (D_free_) was separated by centrifugation (2000 rpm, 5 min) and quantified with HPLC. An encapsulation efficiency (En%) of AP-MVLs was estimated using the following equation.
En(%) = (D_tot_ − D_free_)/D_tot_ × 100%(3)

### 4.8. In Vitro Release and Releasing Mechanism of AP-MVLs

AP-MVL suspensions were diluted with saline solution and centrifugated for 5 min (2000 rpm). The precipitate was well-dispersed in phosphate buffered saline (PBS). Aliquots of the suspension (1 mL) were pipetted into the bag filters (5.0 × 2.5 cm, Mw = 14,000), which was in a flask containing 30 mL PBS (0.01 M). The flasks were incubated at 37 °C under a constant rotation of 100 rpm. Samples were collected at each time point (0.5, 1, 2, 4, 6, 12, 24, 36, 48, 60, and 72 h). Release of the AP solution and a mixture of AP solution and blank MVL suspension was studied the same way. Additionally, common liposome suspensions were prepared using a film dispersion method with the same ratio of drug and lipids, and their release behavior in vitro was also evaluated. The AP concentrations were determined using an UPLC assay.

Aliquots of the suspension (1 mL) were pipetted into the bag filters (5.0 × 2.5 cm, Mw = 14,000), which was in a flask containing 30 mL PBS (0.01 M). The flasks were incubated at 37 °C under constant rotation at 100 rpm. Samples were collected at each time point (0.5, 1, 2, 4, 7, 10, 24, 36, and 48 h), and observed using an optical microscope (×500).

### 4.9. Function Characteristics of AP-MVLs (In Vitro Digestion Protocol)

#### 4.9.1. Thermal Stability

Put peanut peptide respectively at 4, 10, 20, 30 and 60 °C environment, examine the peanut peptide changes in the structure of different temperature, active, polycystic liposomes are stored in 4 at the same time, 37 °C environment, investigate polycystic liposomes in refrigerated storage and transport of stability, ensure the smooth progress of the follow-up experiment digestion.

#### 4.9.2. Simulated Digestion Fluids and Enzyme

Simulated salivary fluid (SSF), simulated gastric fluid (SGF), and simulated intestinal fluid (SIF) were made up of the corresponding electrolyte stock solutions, enzymes (Table 3), CaCl_2_, and water.

In vitro simulated digestion experiments were carried out using an in vitro simulated digestive system (Soochow University) [27,28].

### 4.10. Statistical Analysis

All the experiments were carried out in triplicate and analyzed by a one-way analysis of variance (ANOVA). Data were reported as mean ± standard deviation (SD). Statistical significance of differences was evaluated by Duncan’s multiple range test (*p* < 0.05) using the Statistical Analysis Systems software version 9.0.

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
