# Peer review of "Multivesicular Liposomes for the Sustained Release of Angiotensin I-Converting Enzyme (ACE) Inhibitory Peptides from Peanuts: Design, Characterization, and In Vitro Evaluation"

_molecules, 2019, doi:10.3390/molecules24091746_

Round 1

Reviewer 1 Report

This ms is not well prepared and is very difficult to read and understand. Numerous typos and grammatical errors are seen in everywhere. This ms needs to be edited by native English person.

All figures no legends and without statistical labels, there are hard to read and confirm their results.

What is the reason listed table 1, for what purpose and is also very confused.

L94 Five levels were designed for each factor in the single-factor experiment as shown in Table 2 and 3.

In table, leevls ????

L102 the encapsulation efficiency of AP-MVLs increased to 82.05±2.38% from 71±1.25%. Within the range of 12000 RPM ~18000 RPM, the encapsulation efficiency of the plasmid; what are RPM and plasmid???

L194 At room temperature and 4 centigrade, the leakage rate of 144 h was 21.3% and 59.4% respectively and the drug leakage of liposomal were happened under the two conditions above.

Author Response

First of all, thank you for your careful review of this paper.According to your Suggestions, the following modifications are made:

1. With the help of the editorial department of the journal, carefully revise the normative English language of the full text.

2. As the single-factor experiment was only to select the optimal range for reference of subsequent ball packing model, standard deviation and variance were not marked.

3.RPM was the unit of rotation speed. Meanwhile, plasmid was changed to MLVs.

4. I have carefully sorted out and modified the language according to your tips.

Finally, thank you again for your careful review of this article.

Reviewer 2 Report

Page 2 line 48: define ACE. ACE is defined in the next paragraph but should be defined when first used

Page 2 line 53: insert reference for ACE being beneficial for blood pressure regulation

Page 2 line 68: define “AP”

Page 3 line 90: Is this a subtitle?

Page 3 Table 2 & 3: Please justify the choice of these parameters. Most of them did not show significant difference on the encapsulation efficiency in the range of choice (except he O/W ratio). Authors need to justify why a larger range wasn’t investigated (due to technical constrains or other previous results?). With the current result, it’s hard to tell whether an optimal range/value has been identified or not.

Page 6 line 170-172: It’s hard to tell from the picture that “triglycerides cannot be arranged together with phospholipid composition polycystic skeleton, when organic solvents were removed, but to stay in the chamber junction and in the water phase”. Suggest authors to further elaborate and point out using arrows.

Page 6 section 2.3: show figure of the results

Page 7 line 186: why is the zeta potential value here (70.5+/-2.88) different from page 4 line 151?

Page 8 line 212: should be Fig 5

Page 8 Figure 5: add to figure title to indicate which one is cholesterol which one is trioleate

Page 8 Figure 6, left and middle figures: change x-axis to English

Page 8 Figure 6, right figure: How is “absorption” measured and what does it indicate? Why is this result so much different from Figure 5 and the left and middle figures in Figure 6?

Page 8 Thermal Stability: cannot find the method of obtaining this data. Please elaborate in the Materials and Method Section.

Page 10, section 3: should this be “Conclusions”?

Author Response

First of all, I would like to thank you for your review of this paper and your valuable Suggestions for revision. I have carefully modified this paper according to your Suggestions for revision. The details are as follows:

1. Complete the specification and modification of the full-text language with the help of the editorial department;

2. Standardize the definitions and abbreviations;

3. Based on the existing research and experimental basis, the process parameters are set as the existing range, and the ball stacking model is optimized according to the existing range. The results will be presented in subsequent articles;

4. Triglycerides are marked in the figure and supplemented in the paper;

5. In figs. 5 and 6, labels are added and Chinese is modified into English

6. Figure 5 shows the effects of different concentrations of triglycerides and cholesterol on the encapsulation rate, while figure 6 shows the stability of the polycystic liposome at different temperatures, and absorbance indicates the number of permeated samples

7. During the process optimization, the Zeta potential was the data of blank polycystic liposome, while the data of peanut peptides were included in the following paper. According to your opinion, the data were unified into the data of drug inclusion

Reviewer 3 Report

The Authors here do not provide a critical review of the scientific literature on liposome preparation in hypertension therapy  thus missing some important issues. The data hereby outlined resume an overall weak revision of the recent studies in the field of nanomedicine based therapy.  Please try to better outline the novelty of this investigations  here reported providing a broader and deeper discussion of the recent advances.  The Authors failed to give convincing evidences of the significant public health impact of the here achieved results. Some concerns arise for long term stability and storage of the MLV preparation, a basic issue for oral delivery of long term therapy drugs.This issue should be further discussed. Please try to better outline the novelty of the investigations here reported providing a broader and deeper discussion of the recent advances in the field. The manuscript would need a thorough linguistic revision.

Author Response

First of all, I would like to thank you for your review and Suggestions on this paper. I have modified the whole paper according to your Suggestions on revision:

1. With the help of the editorial department, the full text language has been modified and improved, and now it has been greatly improved;

2. According to your outstanding opinions, some contents were discussed comprehensively and deeply;

3. Combined with experimental data and existing research contents, polycystic liposome has excellent storage stability and is a new in vivo delivery system, providing ideas for drug delivery in vivo in the later stage.

Reviewer 4 Report

In the current manuscript, the authors have designed multi-vesicular liposomes encapsulating peptides derived from peanut meal, characterised the effect of various design parameters on the physicochemical characterization of MLVs and demonstrated their in vitro release and effect in a simulated digestive system.

Could the authors please address the following issues:

Line 24-25: The sentence "The digestion experiment was.." needs to be reformulated

Table 2: Spell check "Levels" 

Line 127: Did the authors not have any dissolution problems with cholesterol when increasing the concentration from 6012mg/mL? Generally, Cholesterol in dissolved at 5mg/mL in methanol/ethanol.

Figure 2 and Figure 3: The legends should be modified and the description of each graph should me mentioned. 

Discussion: This section needs to be expanded and appropriate references must be added.

Line 295, 302: There are some spell checks that needs to be performed.

Methods: Can all the centrifugation speeds be converted from rpm to g values?

Line 346: what the unit of ratio? mL:mL?

Author Response

First of all, thank you for your careful review of this article and put forward valuable Suggestions for revision. I have optimized the article as a whole according to your Suggestions.

1. Revise and improve the English description in the paper with the help of the editorial department.

2. Cholesterol is easily soluble in trichloromethane, and the experimental design factors in this paper can fully dissolve cholesterol.

3. In order to more closely describe the equipment conditions, RPM is directly adopted as the unit of speed.

4. When Zeta potential is tested in this paper, ml is used as the unit.

5. Due to differences in equipment, sufficient guarantee of experimental results and intuitive expression of structure, there is no unified unit.

Round 2

Reviewer 1 Report

The authors should perform the statistical analysis, how many samples were run on each experiment. The statistical significance needs to be labelled and described in figure legends. This work seems to be very preliminary presentaion even this first reversion, but the ahtors did not improve alot.

Author Response

Hello, thank you for your review and guidance of this manuscript. According to your opinions, I have made statistical analysis again and added significant differences.Each experiment was run in three parallel sets and repeated to ensure data preparation.Thanks again for your review